# Trusted Aggregation (TAG): Backdoor Defense in Federated Learning

**Joseph Lavond**           *jlavond@email.unc.edu*
*Department of Statistics and Operations Research*
*University of North Carolina at Chapel Hill*

**Minhao Cheng**           *mmc7149@psu.edu*
*College of Information Sciences and Technology*
*The Pennsylvania State University*

**Yao Li**           *yaoli@email.unc.edu*
*Department of Statistics and Operations Research*
*University of North Carolina at Chapel Hill*

**Reviewed on OpenReview:** *https://openreview.net/forum?id=r9eNUDe2im*

## Abstract

Federated learning is a framework for training machine learning models from clients with multiple local data sets without access to the data in its aggregate. Instead, a shared model is jointly learned through an interactive process between a centralized server that combines locally learned model gradients or weights from the client. However, the lack of data transparency naturally raises concerns about model security. Recently, several state-of-the-art backdoor attacks have been proposed, which achieve high attack success rates while simultaneously being difficult to detect, leading to compromised federated learning models. In this paper, motivated by differences in the logits of models trained with and without the presence of backdoor attacks, we propose a defense method that can prevent backdoor attacks from influencing the model while maintaining the accuracy of the original classification task. TAG leverages a small validation data set to estimate the most considerable change a benign client's local training can make to the shared model, which can be used to filter clients from updating the shared model. Experimental results on multiple data sets show that TAG defends against backdoor attacks even when 40 percent of user submissions to update the shared model are malicious.

## 1 Introduction

Federated learning (FL) is a promising solution for constructing machine learning models from numerous local data sources that cannot be directly exchanged or aggregated (Yang et al., 2019; Kairouz et al., 2021). These limitations become particularly crucial in contexts where data privacy and security are prominent concerns (Li et al., 2020), with healthcare being a prime example. Additionally, FL has garnered significant attention from companies that opt to offload computing workloads onto local devices. Furthermore, FL allows for non-independent and non-identically distributed local data sets. Hence, a shared and robust global model is often unattainable without collaborative learning. Within this FL framework, local entities, called clients, contribute their locally acquired model gradients or weights to be intelligently combined by some centralized entity, the server, resulting in a shared machine-learning model.

However, concerns have arisen regarding the potential vulnerabilities inherent in FL. The lack of control or knowledge of the local training procedures allows malicious users to create updates, compromising the global model for all participating clients. One insidious threat, especially for classification models, is the

targeted backdoor attack, in which malicious actors seek to manipulate the global model into associating specific input data manipulations (known as triggers) with particular outcomes (the target), for example, a particular predicted class label. While various methods have been proposed to detect triggers and defend against backdoor attacks (Kurita et al., 2020; Qi et al., 2020; Li et al., 2021), these approaches often rely on having access to the training data itself, which is not feasible within the FL paradigm. Moreover, the limited information available to detect and prevent such malicious intent within FL makes backdoor attacks more straightforward to execute and more challenging to identify.

In this article, we shed light on an approach to detect and mitigate backdoor attacks in FL. Our primary observation centers on the substantial divergence between the logits of classification models created by malicious users and those produced by benign users. This discrepancy is most pronounced for the class label targeted by the backdoor attack. Leveraging this stark contrast, we propose a method that employs a minimal, clean data set to generate a backdoor-free, locally trained model, which we term the "trusted user". We then compare other user models against this trusted user model on the same clean data to identify models with unusual outputs. Suspicious users are removed from participating in the update of the global model.

To identify unusual outputs, we propose comparing user and trusted models by the distributional difference between their outputs and those of the most recent global model to identify malicious updates. We use the trusted user to estimate the most considerable distributional difference a benign user's local training could produce and eliminate other returning user models that exceed this distance cutoff. We show that robust aggregation is insufficient to defend against backdoor attacks and that our framework is more effective than similar proposed defenses. In FL, multiple attackers can simultaneously attempt to attack the shared model with either the same or different malicious objectives. Our proposed method demonstrates its effectiveness against backdoor attacks with multiple attackers, even when 40 percent of returning updates are part of an attack. Remarkably, our approach consistently outperforms existing methods and remains robust even when backdoor attacks persist throughout federated training. Moreover, our method does not compromise the global model's performance on clean data, ensuring that the accuracy of the original classification task remains intact. We provide experimental evidence across multiple data sets to underscore the consistency and reliability of our results.

The rest of the paper is organized as follows: Section 2 briefly summarizes related work from federated learning, backdoor attack, and defense. We introduce the proposed framework for excluding malicious users from participating in the update of the global model in Section 3. Finally, Section 4 gives empirical evidence for performance improvements over other defense options.

## 2 Related Work

**Federated Learning**  Federated learning (FL) is an emerging machine learning paradigm with great success in many fields (Bonawitz et al., 2019; Hard et al., 2018; Ryffel et al., 2018). At its core, FL operates through iterative rounds of model improvement. In each round, the global model is distributed to participating users, and a subset of these users is selected to update a local copy of the model. These chosen users train their models on their respective local data sets. The resulting models are shared to safeguard data privacy and aggregated to construct a new global model.

**Backdoor Attack**  Recently, the FL setting has become a target for various backdoor attacks. In Xie et al. (2020), the authors highlighted how the multi-user nature of FL could be exploited to create more potent and persistent backdoor attacks. Distributing the backdoor trigger among a few malicious users effectively induced the desired behavior in the global model at higher rates and extended periods after the attack had ceased. Another notable contribution in this domain is the projection method known as Neurotoxin (Zhang et al., 2022). This approach projects the attacker's updates onto dimensions with small absolute weight vector values, claiming that benign users update such weights less frequently, leading to longer-lasting successful attacks. Our research rigorously evaluates our proposed method's effectiveness against both attacks.

**Defense** While FedAvg (McMahan et al., 2017) remains the most prevalent aggregation method in FL, the field has since introduced robust aggregation rules. In particular, Median and Trim-mean, two such methods, were proposed in Yin et al. (2018). The Median operation computes the coordinate-wise median among the weight vectors of selected users. Similarly, the Trim-mean procedure aggregates weight vectors by calculating the coordinate-wise mean after excluding the largest and smallest $k$ elements in that coordinate. Our experiments show that these robust aggregation methods are insufficient against backdoor attacks.

Defending against backdoor attacks in FL has been a relatively unexplored area in the literature. Many previously proposed defenses, such as Neural Cleanse (Wang et al., 2019), are infeasible for FL, as they jeopardize data privacy. While prior work (Shejwalkar et al., 2022) suggested that norm clipping (Sun et al., 2019) could be effective against backdoor attacks in FL, it is vulnerable to the Neurotoxin attack. Other works on federated learning include DeepSight (Rieger et al., 2022), Baffle (Andreina et al., 2020), and FLAME (Nguyen et al., 2021). We focus on comparison with FLTrust (Cao et al., 2020), as it (similar to our proposal) requires an additional small, clean data set. Although the original FLTrust paper showcased its effectiveness against adaptive attacks involving around half of the clients being malicious, our research demonstrates its failure even against the most basic backdoor attacks considered in our experiments. Thus, our work addresses a critical gap in federated learning and enhances similar defense methodologies.

## 3 Trusted Aggregation (TAG)

This section begins with an observation in Section 3.1 that model performance on clean data changes depending on whether the model was backdoor attacked. Next, Section 3.2 introduces our framework for excluding users from updating the global model depending on their local model's performance on some clean data. Finally, Section 3.3 provides a smoothing procedure to strengthen our defense further.

### 3.1 Motivation

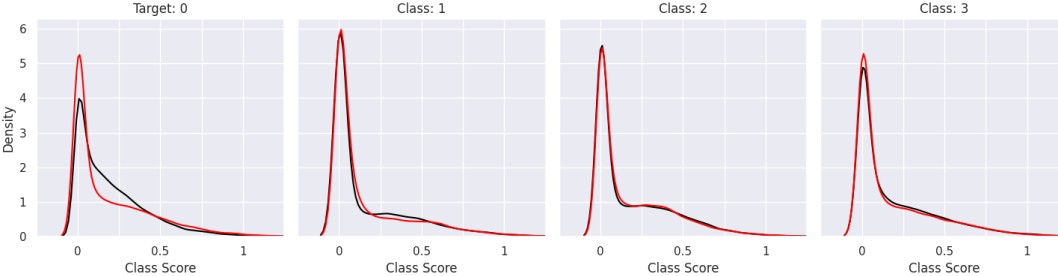

Figure 1: Output distributions (kernel density estimation based) conditional on the class label for a backdoor model (black) and a clean model (red). Note the obvious difference between the distributions of the backdoor and clean models for the target label class.

Our proposed method, Trusted Aggregation (TAG), is motivated by the observation that the distributions of model logits generated by malicious users significantly differ from those produced by benign users. We use the term logits to refer to the output of a classification model before the softmax operation produces probabilities. Note that each element of the logit vector corresponds to precisely one class, establishing a unique node-class association for the logits.

In targeted backdoor attacks, malicious users aim to create an additional learned association between a particular manipulation of input data (the trigger) and a specific class label (the target). The additional learned association effectively transforms the task into a $(m + 1)$-way classification problem. We intuitively believe this difference in the model's task can be exploited to identify models trained with a backdoor attack. In Figure 1, we demonstrate that this learned association can lead to a distributional change in the logits on non-attacked inputs, particularly for the target class. Our insight suggests that models containing a backdoor may produce distinct distributions of logits when provided with clean data. Consequently, if we

have a model believed to be trained without a backdoor attack, we can identify whether another candidate model exhibits signs of a backdoor attack by comparing their logits on the same clean data.

## 3.2 Detection Framework

Our detection framework assumes the presence of a small, clean validation data set, which serves as the gatekeeper for updates to the global model. This data set can either belong to a trusted existing user or be collected by the centralized server and treated as a new user. We will refer to this trustworthy validation data set as the "trusted user".

Our detection method leverages the trusted user to evaluate incoming model weights and determine whether each contribution can participate in the global model update process. The core idea is to detect user models with unusually distributed outputs using a clean data set from the trusted user. Our method can be easily extended when multiple trustworthy data sets or users are available. Refer to Figure 2 for an overview of our proposed detection framework.

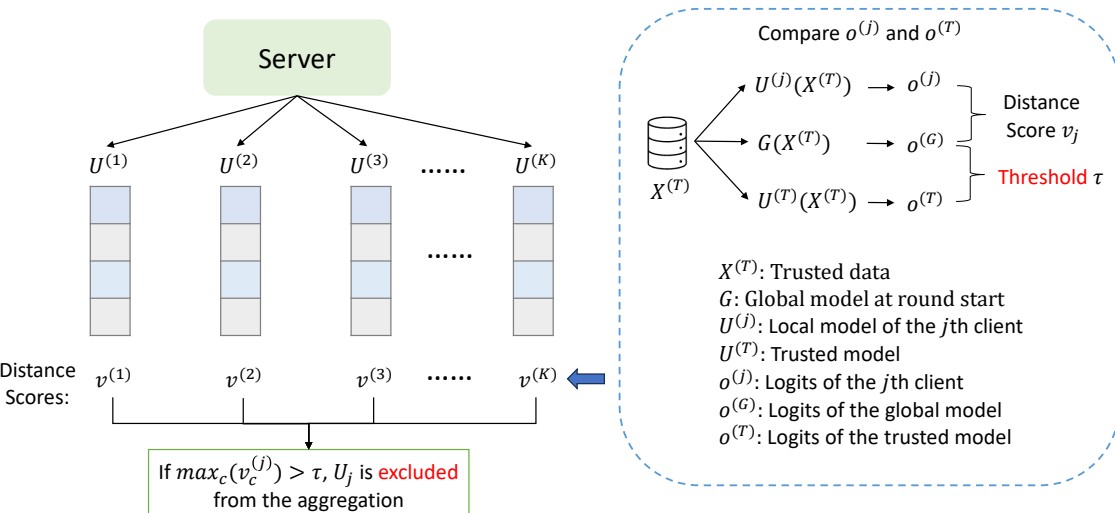

Figure 2: Diagram representation of our trusted aggregation detection framework. A distance score ($\mathbf{v}^{(j)}$) is calculated for each selected user's model ($U^{(j)}$) based on the distributional distances between the user's model and the global model. A threshold ($\tau$) is computed based on the distributional distances between the trusted user's model ($U^{(T)}$) and the global model ($G$). If the distance score of a user is greater than the threshold, it will be excluded from the aggregation. See more details in Section 3.2.

In each communication round, the trusted user performs the following steps to establish a threshold for detecting malicious users:

1. The validation data is utilized to update a copy of the current global model simultaneously with the local training of other users.

2. When models are returned by the subset of users selected to participate in the shared model update potentially, logits are generated and stored for the validation data. These logits are denoted as $\mathbf{o}^{(G)}$, $\mathbf{o}^{(T)}$, and $\mathbf{o}^{(j)}$ for the global, validation, and $j$-th user models, respectively.

3. Subsequently, we compute the distance between each user from the current global model. Specifically, for each class $c$, we calculate the class-conditional distributional distance $\mathcal{D}$ between empirical distributions for the current global model's logits and the user's logits using some distributional difference function.

4. Given $m$ total classes, the process generates a distance vector $\mathbf{v}$ for each user, including the trusted user. These distance vectors then dictate which users can participate in the model. See Algorithm 1 for additional details.

---

**Algorithm 1** Trusted Aggregation (TAG)

Let $\mathcal{S}$ denote the subset of users selected to update the global model for a given round $r$.

---

**Input:** Global model $G$, user models $\{U^{(j)}\}_{j \in \mathcal{S}}$, trusted model $U^{(T)}$ and data $\mathbf{X}^{(T)}$, and scaling coefficient $\theta \geq 1$.

1: Generate logits $\mathbf{o}^{(G)} = G(\mathbf{X}^{(T)})$, $\mathbf{o}^{(j)} = U^{(j)}(\mathbf{X}^{(T)})$ for $j \in \mathcal{S}$, and $\mathbf{o}^{(T)} = U^{(T)}(\mathbf{X}^{(T)})$

2: **for** Each class $c \in [1, \ldots, m]$ **do**

3:    Compute distributional distances between each user and the global model $\mathbf{v}_c^{(j)} = \mathcal{D}\left(\mathbf{o}_c^{(j)}, \mathbf{o}_c^{(G)}\right)$ for $j \in \mathcal{S}$ and $\mathbf{v}_c^{(T)} = \mathcal{D}\left(\mathbf{o}_c^{(T)}, \mathbf{o}_c^{(G)}\right)$

4: **end for**

5: Compute threshold for communication round $r$
$\hat{\tau}_r = \theta \times \max_c \left[ \mathbf{v}_c^{(T)} \right]$

6: Exclude suspicious users
$\mathcal{S}_r = \left\{ j \in \mathcal{S} \mid \max_c \left[ \mathbf{v}_c^{(j)} \right] \leq \hat{\tau}_r \right\} \subseteq \mathcal{S}$

7: **return** Update $G$ with $\{U^{(j)}\}_{j \in \mathcal{S}_r}$

---

We aim to estimate the maximum change a non-malicious user can introduce in communication round $r$, defined below in Equation 1:

$$\tau_r = \max_j \left( \max_c \left[ \mathcal{D}\left(\mathbf{o}_c^{(j)}, \mathbf{o}_c^{(G)}\right) \right] \right) \tag{1}$$

Users with distance values surpassing this threshold should be excluded from the update process. For estimation of $\tau_r$, we compute $\max_c \left[ \mathcal{D}\left(\mathbf{o}_c^{(T)}, \mathbf{o}_c^{(G)}\right) \right]$ for our trusted user. Note $\tau_r$ involves the maximum of all benign users. Since the validated user is non-malicious, their distance vector serves as a good representation of other non-malicious users. However, we scale by $\theta \geq 1$ since the actual maximum will be at least as large as our observed trusted user. A user with a maximum distance smaller than the threshold $\hat{\tau}_r = \theta \times \max_c \left[ \mathcal{D}\left(\mathbf{o}_c^{(T)}, \mathbf{o}_c^{(G)}\right) \right]$ is considered a benign user. In comparison, a user with a maximum distance larger than or equal to the threshold will be removed. However, this naive threshold is precarious, and due to its instability, a lucky malicious user can get past it in some rounds. To overcome this limitation, we propose a specific smoothing procedure in Section 3.3

If the distributions of $\mathbf{v}_j = \mathcal{D}\left(\mathbf{o}_c^{(j)}, \mathbf{o}_c^{(G)}\right)$ can be assumed, the ranges of plausible $\theta$'s can be better determined.

**Proposition 1.** *If $v_c \sim Uniform(0, b_c)$ for all classes $c \in [1, \ldots, m]$ and for all benign users, then $\mathbb{E}[v] \leq b \leq E[2v]$ where $v = \max_c [v_c]$ and $b = \max_c [b_c]$.*

For example, when $\mathbf{v}_c^{(j)}$ are each uniformly distributed for all users, Proposition 1 suggests that, on average, $\hat{\tau} = \theta \times \max_c \left[ \mathbf{v}_c^{(T)} \right]$, should equal $\tau = b$ for some $\theta \in [0, 1]$. Yet, we acknowledge that it may be unreasonable to assume that class conditional distances are Uniform as many training hyper-parameters and even model choice will impact the distance distributions. A better approach could be to allow the data to determine the scaling factor, i.e., selecting the scaling factor based on the distribution of logits observed in the experiments. However, in our experimental results in Section 4, we often found that simply setting $\theta = 2$ outperforms existing methods, so we did not further explore data-dependent scaling factors. See Section B.2, for a sensitivity analysis on how our scaling coefficient impacts model performance for non-iid users. We present these results to assist in understanding reasonable magnitudes for $\theta$ as many distributions may not require large scaling values.

We recommend choosing $\theta$ based on the setting's prevalence of backdoor attacks (potentially unknown) and the cost of a successful attack to interested parties. Relatively large values for $\theta$ will allow more users to update the global model but increase the risk of a successful backdoor attack. Conversely, using our threshold $\hat{\tau}_r$ without scaling would help to prevent stronger backdoor attacks, but with the potential loss of denying benign users from updating the global model. The goal is to choose the smallest $\theta$ that allows benign users to create a robust shared machine learning model sufficiently. For a more substantial discussion on the fairness of our algorithm, please see Section B.1.

### 3.3 Global-Min Mean Smoothing (GMMS)

Stabilizing the threshold value is essential for maintaining the security of our method. While a straightforward approach to achieving this stability is through a smoothing technique, such as a moving average, it comes with challenges. The naive threshold value fluctuates in the early communication rounds as the model learns to relate inputs and output classes quickly. We need to understand when past behavior of $\hat{\tau}_r$ is relevant while ensuring stability throughout the process.

Conventional smoothing methods, which rely on several previous values, can lead to an overly high threshold in the initial rounds. A falsely high cutoff, in turn, could create vulnerabilities that attackers could exploit. To address this concern, we introduce our Global-Min Mean Smoothing approach, which combines the benefits of both a stable threshold and a rapidly adjusting threshold early in training.

The foundation of our approach lies in using the lowest observed value of $\hat{\tau}_r$ (Global Min) as the starting point for the (Mean) smoothing window. Here, $\hat{\tau}_r$ represents the naive threshold estimation up to round $n$. The smoothed threshold $\tilde{\tau}_n$ for round $n$ is determined using Algorithm 2.

---

**Algorithm 2** Global-Min Mean Smoothing (GMMS)

---

**Input:** Threshold history $\hat{\tau}_1, \ldots, \hat{\tau}_r$

1: Identify the start of the smoothing window
$$s = \arg\min_{i=2,\ldots,r} = \hat{\tau}_i$$
2: **return** Smooth estimate $\tilde{\tau}_r = \dfrac{1}{r-s+1} \sum_{i=s}^{r} \hat{\tau}_i$

---

In scenarios where $\hat{\tau}$ exhibits rapid initial decreases, we should observe new global minimums, which serve as a reset point for the threshold smoothing. Starting our smoothing from the global minimum allows us to maintain the original threshold sequence's decreasing behaviors. However, when the original sequence is not experiencing a decline, previous values are utilized to smooth the threshold, effectively preventing lucky malicious users from evading a volatile threshold.

Figure 3 visually compares our Global-Min Mean Smoothing with the base (naive) threshold and various conventional smoothing methods. It is evident that our Global Min-Mean Smoothing not only captures the early behavior of the naive threshold but also significantly improves stability, rendering it a robust choice for safeguarding the global model against backdoor attacks.

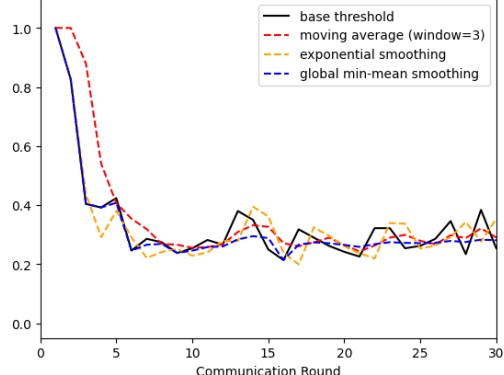

Figure 3: Comparison of the global min-mean smoothing with the naive threshold and various smoothing methods.

## 4 Experiments

In this section, we present a series of experiments that offer valuable insights into the effectiveness of Trusted Aggregation (TAG) as a defense mechanism against strong backdoor attacks in federated learning. In

Section 4.2, we observe that robust aggregation is insufficient to prevent backdoor attacks and that TAG outperforms the backdoor defense FLTrust, which also requires additional clean data. Additionally, Section 4.4 shows that the TAG is robust to changes in the data distribution of clients or the trusted data set.

## 4.1  Setting

This section provides a comprehensive breakdown of the parameters used throughout our study to ensure the reproducibility of our experiments, as summarized in Table 1. When used with the code available on our GitHub repository [1], this detailed information serves readers seeking to recreate our results and further investigate our work.

|  | Hyperparameter | Variable Name | CIFAR10 | CIFAR100 | STL10 |
|---|---|---|---|---|---|
| Federated Learning | Users | $n\_users$ | 100 | 100 | 20 |
|  | Local Data Size | $n\_user\_data$ | 500 | 500 | 400 |
|  | User Subset Proportion | $p\_report$ | .1 | .1 | .5 |
| Data Augmentation | Padding |  | 4 | 4 | 12 |
|  | Random Horizontal Flip | NA | .5 | .5 | .5 |
|  | Random Crop Size |  | 32 | 32 | 96 |
| All Users | Batch Size | $n\_batch$ | 64 | 64 | 128 |
|  | Weight Decay | $wd$ | 5e-4 | 5e-4 | 5e-4 |
| Benign Users | Local Epochs | $n\_epochs$ | 10 | 10 | 10 |
|  | Learning Rate | $lr$ | .01 | .01 | .01 |
| Malicious Users | Local Epochs | $n\_epochs\_pois$ | 20 (15) | 15 | 25 (15) |
|  | Learning Rate | $lr\_pois$ | .01 | .01 | .005 (.01) |
| Data Poisoning | Poisoning Proportion | $p\_pois$ | .1 | .1 | .1 |
|  | Stamp Pixel Height | $row\_size$ | 4 | 4 | 24 |
|  | Stamp Pixel Width | $col\_size$ | 4 | 4 | 24 |
| Backdoor Defense | TAG Scaling ($\theta$) | $d\_scale$ | 2 | 2 | 1.1 |
|  | Trim Mean | $beta$ | .2 | .2 | .2 |

Table 1: Default arguments for all experiments unless otherwise specified. For all experiments, alternative values for $\beta$ did not prevent the backdoor attacks. Any values modified for Neurotoxin attacks are shown in parentheses.

**Model**  Our experiments employ the ResNet18 model (He et al., 2016), a well-established classifier. Additionally, to showcase the robustness and generalizability of our approach, we reproduce the main results using the VGG16 model (Simonyan & Zisserman, 2014) in Section A.2. Importantly, we assume that all users, including potential malicious actors, have complete control over various aspects of local training. For simplicity, we use two sets of hyper-parameters for benign and malicious users. The malicious users will poison (add their backdoor trigger) and change the training label to the target class for a given proportion of their local data. They intend their model to associate the trigger with the target class and transfer such behavior to future global models.

**Attack and Defense**  To assess the effectiveness of TAG, we operate in a scenario where the backdoor attack is particularly potent. We mandate that the *same* set of malicious users are included *every* round in the subset of selected users responsible for updating the global model. Moreover, all attacks start in the first communication round. This approach circumvents the randomness associated with selecting users, allowing malicious users to influence the global model repeatedly. Furthermore, the guaranteed benign validation user is excluded from participating in global model updates. These decisions are made to showcase TAG's ability to thwart even backdoor attacks against the shared model in the most substantial attack settings.

---

[1]https://github.com/JoeLavond/TrustedAggregation

In our experiments, we compare the performance of the TAG method with robust aggregation methods, including Median and Trim-mean (Yin et al., 2018), as well as the backdoor defense FLTrust (Cao et al., 2020). Regarding TAG, in our experiments, we exclusively use the Kolmogorov-Smirnov (KS) distance between distributions because it is easy to compute for estimated cumulative distribution functions. However, we acknowledge that other distance functions, and even divergences, may be suitable. We assess these methods against two state-of-the-art backdoor attacks in federated learning: Neurotoxin (Zhang et al., 2022) and Distributed Backdoor Attacks (DBA) (Xie et al., 2020). To evaluate the strength of the considered defenses, we vary the proportion of malicious updates from 10, 20, and 40 percent of the selected users.

**Data**   The experiments are conducted on three datasets: CIFAR10 and CIFAR100 (Krizhevsky & Hinton, 2009), and STL10 (Coates et al., 2011). A specific adjustment pertains to our use of STL10, where we have inverted the conventional train and test data splits. Since we are using only labeled data, we swap train and test to have the larger of the two for training purposes. User datasets are constructed by random sampling from the training data splits. These samples are generated using a Dirichlet distribution that determines the class frequency. In our experiments, we ensure the creation of balanced local data sets by applying a scaling factor, $\alpha = 10000$, to a vector consisting of ones, with the dimensionality equal to the total number of classes. When investigating imbalanced user (and trusted) data sets, we modify $\alpha$ to a value of 1 to create the desired imbalance. Further details, including experiment specifics and hyperparameters, can be found in Table 1.

We split the test set into two parts to evaluate the global model. The first half is used to determine classification accuracy. In the second half, we introduce the backdoor trigger to the images, remove any observations related to the target class, and report the attack success rate. The attack success rate measures the extent to which the backdoor attack has compromised the model by determining the proportion of the poisoned half predicted as the target class. An effective defense method will exhibit a low attack success rate while maintaining a high classification accuracy, indicating that the attack is unsuccessful and the defense does not negatively impact classification performance.

### 4.2   Comparison of Defense Methods Against Backdoor Attacks

To thoroughly evaluate TAG's performance, we explore settings where 10, 20, and 40 percent of the returning user models are malicious in each communication round. Figure 4 provides a visual representation of the performance of various methods against backdoor attacks on three different data sets. We assess the success of each method in terms of attack success rate while ensuring that classification accuracy, Figure 8, remains high. For our primary results, we use scaling coefficients ($\theta$) of 2, 2, and 1.1 for the CIFAR10, CIFAR100, and STL10 data sets, respectively. Our findings reveal that TAG effectively neutralizes the backdoor attack in each case without significantly compromising the classification accuracy of the original task.

Other methods, such as coordinate-wise Median, Trim-mean, and FLTrust, fail to thwart backdoor attacks, with or without Neurotoxin, at all considered strength levels. FLTrust, while capable of delaying attack success in some settings, ultimately falls short in preventing backdoor attacks. The critical difference between TAG and the baseline methods is that while the baseline approaches differentiate between malicious and benign users based on update gradients, our approach compares task performance based on model outputs. For backdoor attacks, the loss is typically a combination of the original task loss and the backdoor loss: Loss = Original Task Loss + $\lambda \times$ Backdoor Loss, where $\lambda$ is usually close to zero. As a result, the gradients of malicious users may appear similar to benign users, but our method can still detect differences between the resulting models. However, different local minima can produce similar model outputs in highly non-convex loss landscapes. In such cases, our method may not be as effective at filtering out models that are more easily detected by gradient-based defenses. Yet, in none of our experimental settings were gradient-based defenses successful in defending against any of the attacks we considered. We remark that our method could filter out suspicious users before other defenses, making it possible to combine with other strategies further to enhance the robustness of models against targeted backdoor attacks.

Our supplemental experiment, detailed in Section A.1, supports that TAG does not hinder performance for the original classification task, even without backdoor attacks. This comparison with FedAvg highlights the minimal impact on the classification task's performance when using TAG as a defense mechanism.

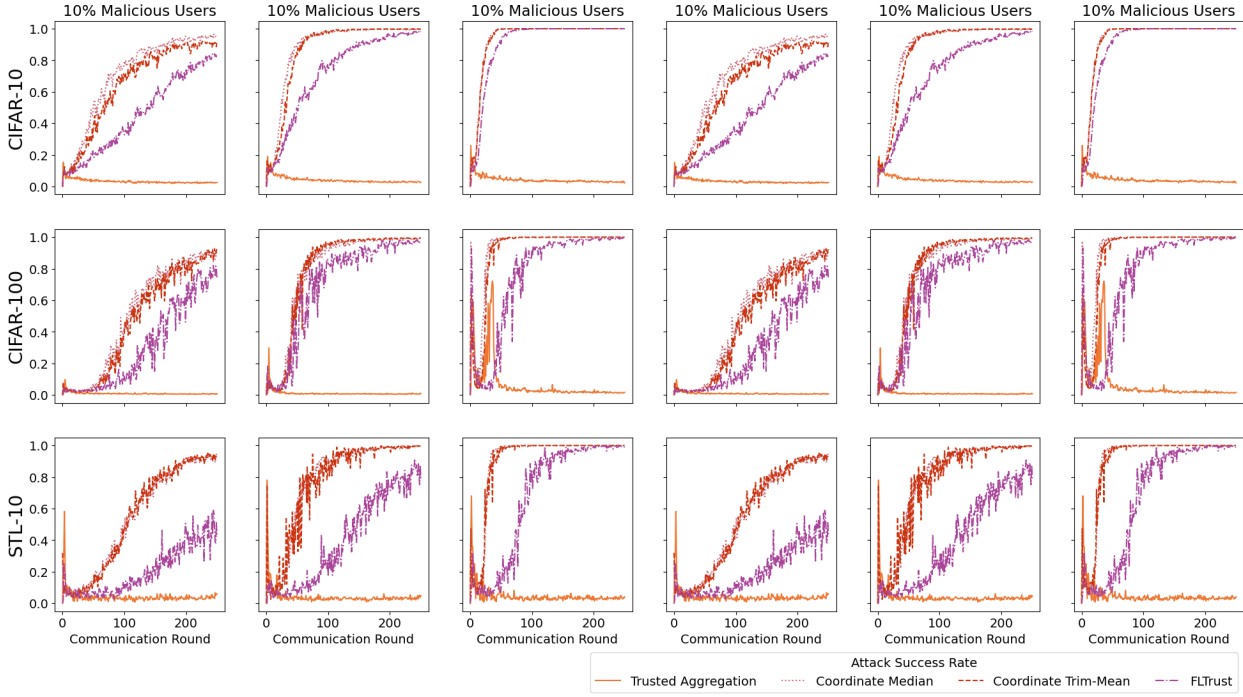

Figure 4: Model performance under DBA without and with Neurotoxin (NT) with 10, 20, and 40 percent malicious updates on several data sets. Column names indicate attack settings, while rows correspond to data sets. The proposed method, TAG, performs well in defending against backdoor attacks as the attack success rates are low. The other methods do not work well against any backdoor attacks.

These results emphasize that TAG enhances model security without negatively affecting the classification task, even without an attack. Furthermore, Section A.2 demonstrates that our primary results, which were obtained using ResNet18, are not model-dependent by reproducing them using VGG16. In conclusion, Trusted Aggregation (TAG) is a crucial advancement in bolstering model security within the federated learning framework.

### 4.3 Necessity Of Threshold Smoothing

We revisit the last attack on the STL10 data set to highlight the importance of our proposed global-min mean smoothing technique. Recall that our smoothing is intended to improve the stability of our estimated threshold while preserving its behavior in the initial rounds, which is not conserved by other smoothing techniques. Suppose we repeat the backdoor attack, with 40 percent of the user subset being malicious each iteration, and omit the global-min mean smoothing. In that case, our method can no longer prevent the backdoor attack with Neurotoxin projection on STL10. Please see Figure 5. We do remark that this was the only attack in our main results that became successful without including smoothing. Regardless, without smoothing, we conclude that malicious users may be able to get past a less stable cutoff.

### 4.4 Extending Results To Imbalanced User Data Sets

In federated learning, user data sets may not adhere to the assumption of being independent and identically distributed. This section explores TAG's effectiveness with imbalanced user data sets, specifically focusing on the CIFAR10 data set. We investigate this scenario under the most potent attack setting, where 40 percent of user submissions are malicious in each round. In Section B, we include our analysis on the weaker attack settings, revealing that TAG remains highly effective in defending against backdoor attacks, even without fine-tuning the scaling coefficient ($\theta$). These results are consistent across imbalanced user data sets, irrespective of whether the trusted data set is imbalanced.

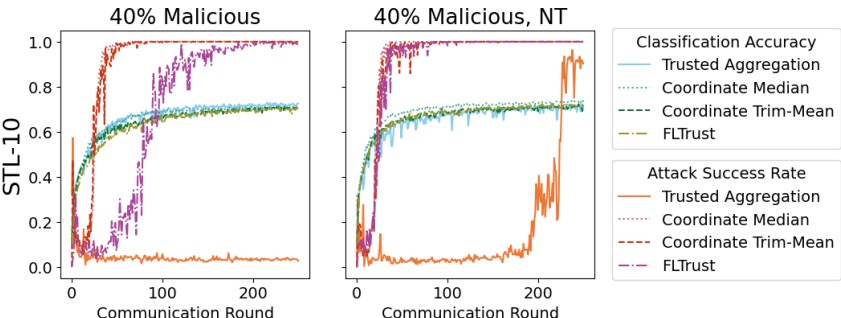

Figure 5: Model performance under DBA without and with Neurotoxin (NT) with 40% malicious updates on STL10 without using global-min mean smoothing. Column names indicate attack setting. TAG fails to defend against the backdoor attack with Neurotoxin without our proposed smoothing technique for improved stability.

In these experiments, we employ the $m$-dimensional Dirichlet distribution with parameter $\alpha\mathbf{1}_m$ to determine the proportion for each class label in the data set, introducing an imbalance. The choice of $\theta$ plays a critical role in the performance of TAG. With appropriate tuning, TAG can successfully neutralize 40% of malicious backdoor attacks. See Figure 6. However, a trade-off exists between backdoor attack prevention and original classification task performance. Smaller scaling values may reduce classification accuracy in the early communication rounds as fewer users contribute to model updates. Nonetheless, as the global model converges, TAG's round-to-round accuracy stabilizes and matches or surpasses the baseline methods. In conclusion, TAG can mitigate backdoor attacks on imbalanced data without compromising the accuracy of the original task, underscoring its utility in a wide range of federated learning applications.

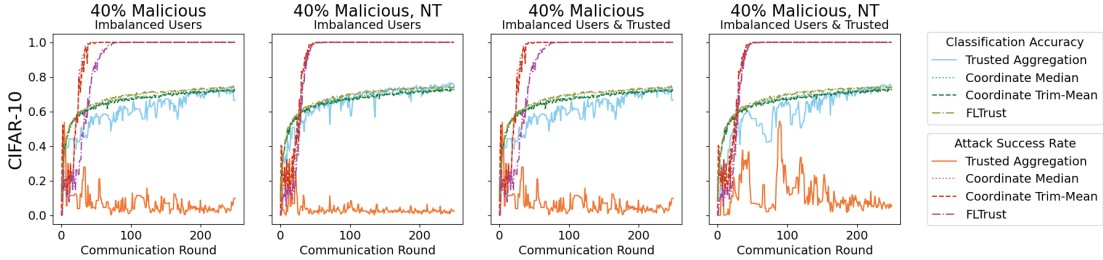

Figure 6: Model performance under DBA without and with Neurotoxin (NT) backdoor attacks with 40% malicious updates on CIFAR10 under imbalanced local data sets with tuned $\theta$. Column names indicate whether the trusted user (Trusted) is also imbalanced. The proposed method, TAG, performs well against backdoor attacks, even when the local user data sets are imbalanced. Again, the other defense methods do not prevent any backdoor attack under imbalanced data.

When obtaining a balanced, trusted data set is challenging, TAG still exhibits its efficacy. TAG effectively prevents backdoor attacks even when the trusted user's data set is imbalanced, and 40 percent of user updates are malicious. These experiments reaffirm TAG's ability to serve as a potent defense mechanism, regardless of the balanced representation within the trusted data set. TAG remains a powerful tool for backdoor defense, even when obtaining a balanced, trusted data set proves difficult. Furthermore, Section B illustrates that significantly reducing the size of the trusted data set relative to other users also does not impact the success of TAG as a defense method. Even when FLTrust operates with a full-sized, representative, trusted user data set, it is outperformed by TAG. These findings underscore TAG's robustness and versatility in handling backdoor attacks in federated learning, making it a valuable addition to the security toolkit.

# 5 Limitations

While Trusted Aggregation (TAG) demonstrates substantial success as a defense mechanism against backdoor attacks in federated learning, several limitations must be considered:

**Extreme Non-IID Distribution** Our experiments do include non-independent and non-identically distributed (non-IID) data sets. However, more extreme cases of non-IID data sets remain unexplored. For instance, we do not test TAG in scenarios where each user possesses the entirety of only a few of the total classes. In such cases, the assumption that this work's primary objective is to learn a single shared model for all users may not be reasonable.

**Limited Application Scope** While our framework is conceptually extendable to various machine learning models, such as regression and natural language processing (NLP), our experimental results are confined to standard classification computer vision models and databases. Future work may be needed to adapt our method and demonstrate its effectiveness in other applications.

We do not claim that TAG can successfully defend against adaptive attacks. Adaptive attacks occur when malicious attackers know our defense is in use. They strive to return a model with output distributions similar to those of a benign model while still exhibiting backdoor behavior, which can be achieved by creating two copies of the global model: one trained on original, unmodified local data to produce a benign model (copy A) and the other (copy B) backdoor attacked with the poisoned data set. The attacker enforces similarity between the outputs of these two copies using techniques like L2 regularization as shown in Equation 2

$$\mathcal{L}(x, y) = \mathcal{L}(B(x), y) + \frac{\mu}{2}||B(x) - A(x)||_2^2 \tag{2}$$

Regardless, TAG outperforms other considered defense methods, even under adaptive attack. See Figure 7. However, the effectiveness of TAG diminishes when faced with strong adaptive attacks.

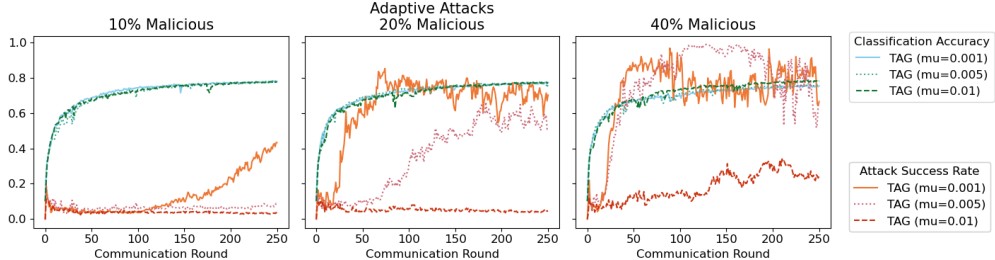

Figure 7: Comparison of TAG under adaptive attack with baseline defense methods not under adaptive attack for various attack settings on CIFAR10. Adaptive attacks are successful against TAG and are a limitation of our proposed defense. However, even under adaptive attack, TAG exhibits lower attack success rates at termination than baseline defenses, not under adaptive attack.

**Adaptive Attacks** While TAG offers a valuable defense mechanism for backdoor attacks in federated learning, its efficacy may be compromised in specific scenarios and attack types. These limitations should guide further research in the field of federated learning security.

# 6 Conclusion

In this study, we introduced Trusted Aggregation (TAG), a robust defense mechanism against backdoor attacks in the federated learning framework. Our extensive experimentation and analysis have led us to several key conclusions:

**Defense Efficacy** TAG has proven a highly effective defense method, showcasing its ability to thwart state-of-the-art backdoor attacks in challenging settings. While existing defense techniques falter under mild attack conditions, TAG consistently prevails against powerful adversaries.

**Heterogeneous Data** TAG's adaptability and resilience are evident in federated learning scenarios involving users with heterogeneous and imbalanced data. Importantly, it accomplishes this without sacrificing classification accuracy for the original task, a testament to its versatility and broad applicability. Furthermore, our method's performance remains stable regardless of the size or distribution of the trusted data it relies on.

**Compatibility** TAG can be seamlessly integrated with other filtering methods or modifications to the aggregation process, enhancing its compatibility with a wide range of defense strategies. This flexibility empowers federated learning systems to adopt a multi-layered approach to security, further safeguarding against adversarial threats.

In summary, TAG is a pivotal advancement in model security for federated learning. It raises the bar for similar defenses against backdoor attacks with its adaptability, robustness, and potential as part of a holistic defense strategy. As federated learning continues to evolve, TAG represents a valuable tool for maintaining model integrity and trust in collaborative machine-learning environments.

# 7 Acknowledgments

We thank the anonymous reviewers and the Action Editor for their constructive feedback, which improved this work. This work was partly supported by the National Science Foundation under grants DMS-2152289, DMS-2134107, and the Cisco Faculty Award.

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

## Appendix

## A   Comparison of Defense Methods Against Backdoor Attacks (Continued)

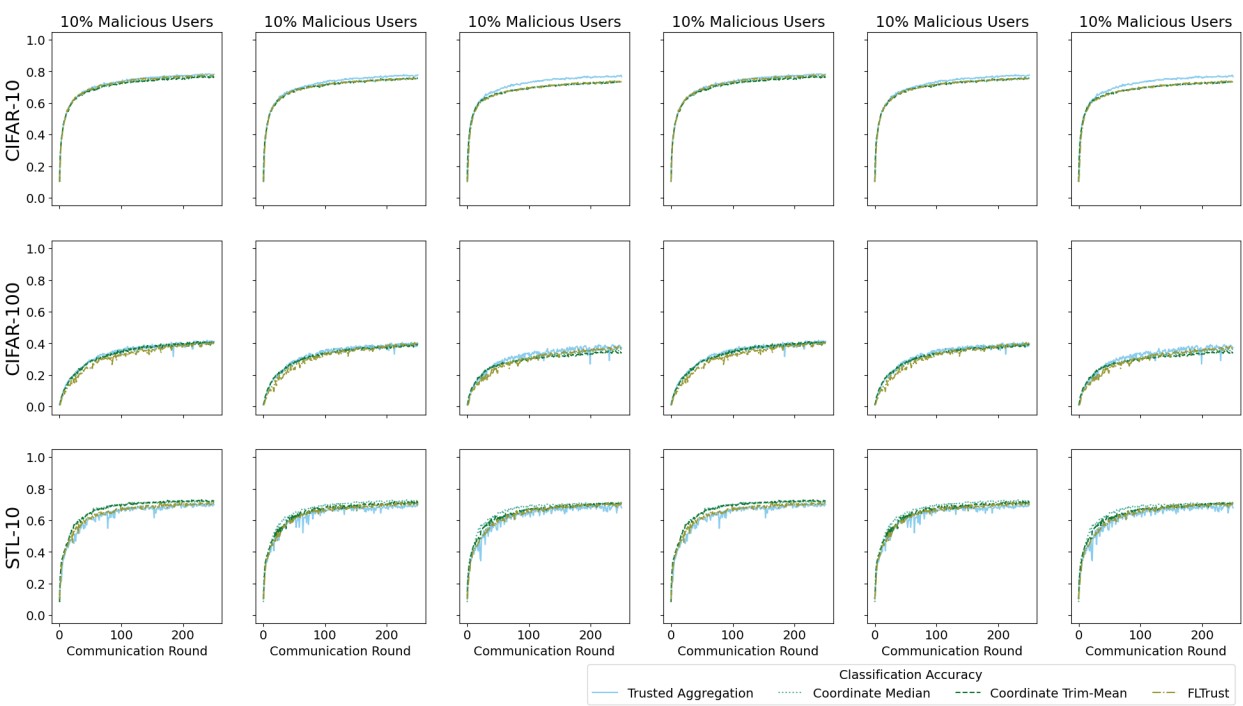

Figure 8: Model performance under DBA without and with Neurotoxin (NT) with 10%, 20%, and 40% malicious updates on several data sets. Column names indicate attack settings, while rows correspond to data sets. All methods result in similar classification accuracies, indicating that TAG offers improved backdoor defense without cost to original task performance.

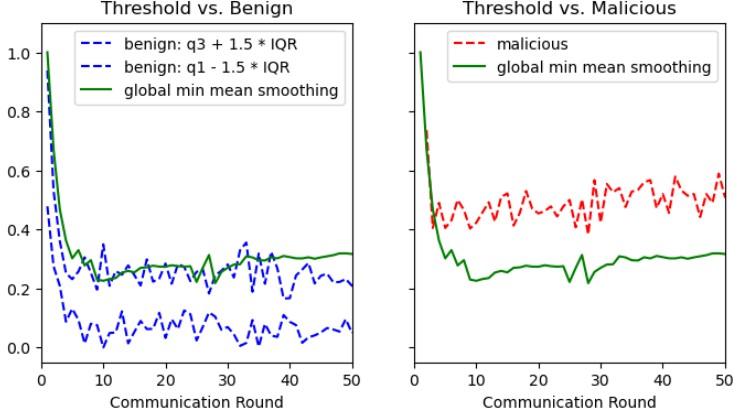

Figure 9: Global Min Mean Smoothing threshold visualized for the first 50 communication rounds with 10 percent malicious updates on CIFAR10. Our threshold can easily differentiate between benign and malicious updates, and almost all benign updates contribute to the global model. Similar results hold under different data sets and attack settings.

## A.1 TAG Classification Accuracy Without Attackers

A successful backdoor defense can prevent attackers while best preserving the model's performance on the original task. The model ability for the original task must also be maintained in the absence of an attack. In addition to successfully preventing attacks when present, we observe in Figure 10 that our defense does not hinder the classification accuracy of the original task on STL10 compared to the FedAvg procedure. The lack of performance degradation in the absence of attacks supports the overall usefulness of our proposed method. The shared model will improve security with our defense without cost if attacks do not threaten the system.

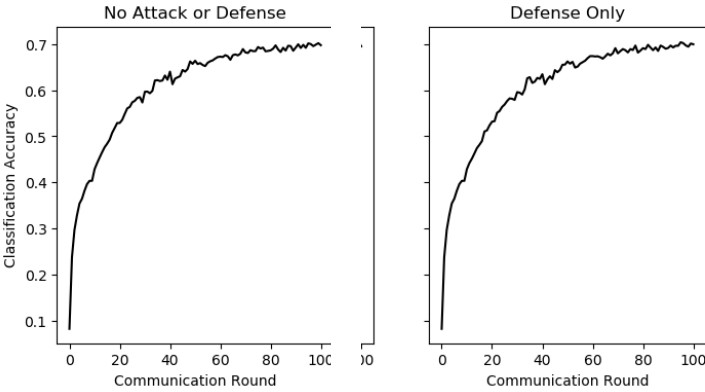

Figure 10: Model classification accuracy on STL10 in the absence of backdoor attacks without (*left*) and with (*right*) TAG defense.

## A.2 Model Generalizability

In this section, we demonstrate that our results are not architecture-dependent by repeating our main experiment results on CIFAR10 for another off-the-shelf image classification model. The following results are obtained using VGG16 with batch normalization, originally proposed in Simonyan & Zisserman (2014). Trim Mean is parameterized by $\beta = 0.1$ in this experiment. However, other values for $\beta$ did not impact defense.

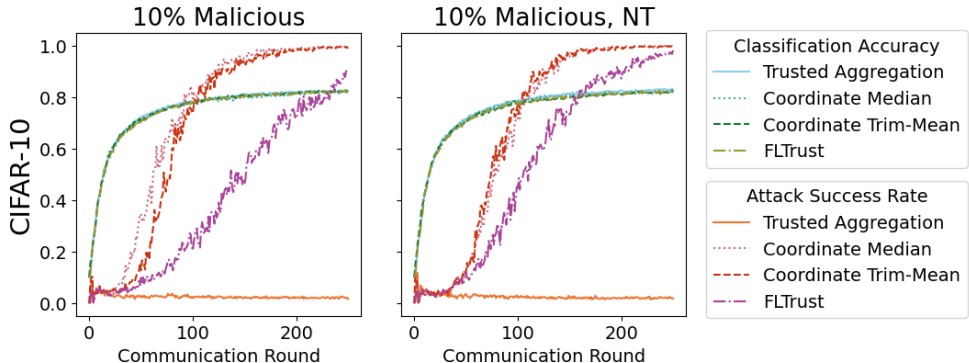

Figure 11: VGG model performance under DBA without and with Neurotoxin (NT) with 10% malicious updates on CIFAR10. Column names indicate attack setting. Results are the same as when using ResNet18. TAG is the only method to prevent backdoor attacks, as shown by low attack success rates while maintaining desirable classification accuracy.

Recall that we attempt to choose the smallest scaling coefficient $\theta$ such that the global model obtains desirable classification accuracy. For CIFAR10, $\theta = 2$ is successful for the ResNet18 architecture both for preventing backdoor attacks and for good classification accuracy for the original task. However, VGG required $\theta = 2.5$ to train a sufficient global model under Neurotoxin attacks. We observe that the choice of $\theta$ may depend on many hyperparameters of various parts of the federated learning procedure. In all attack settings, as presented in our main results, Section 4.2, our proposed defense TAG is the only method successful in preventing backdoor attacks. Again, we note that the successful defense of TAG is not associated with a meaningful change in global model performance compared to the original classification task. We conclude that TAG can be an effective backdoor defense for federated learning for various model choices.

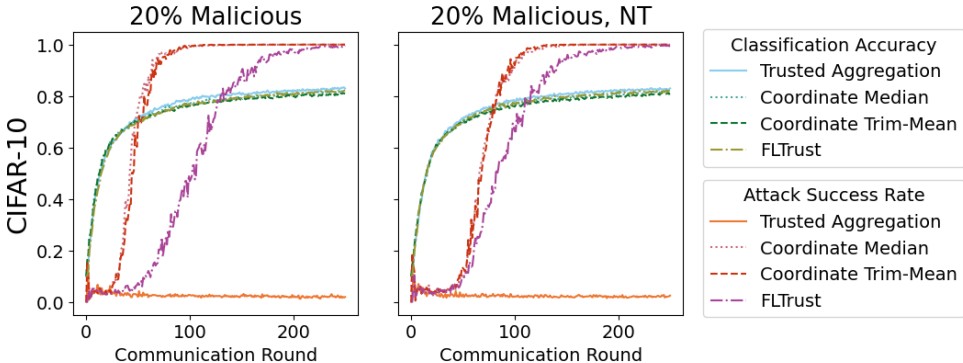

Figure 12: VGG model performance with 20% malicious updates. See above for further details.

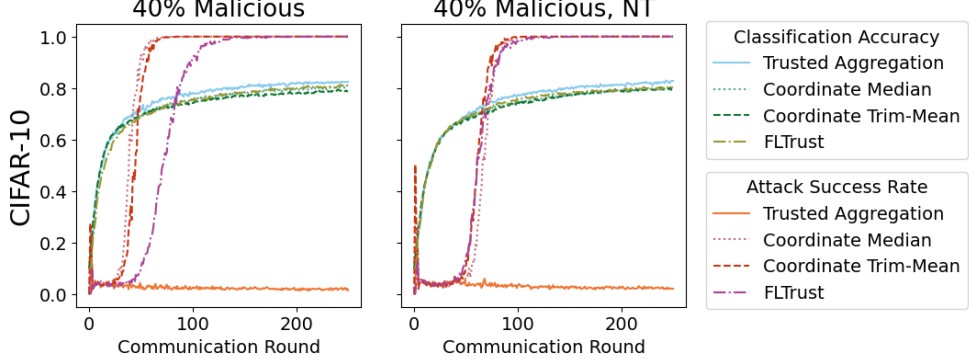

Figure 13: VGG model performance with 40% malicious updates. See above for further details.

## A.3   Size of Trusted Data Set

Additionally, we want to determine whether our method depends on the size of the trusted data set. Hence, we revisited the most substantial attack setting for the CIFAR10 data set but only allowed the validation user to have a data set 20 percent the size of the other local users'. Note that for this experiment, all users have balanced and representative data. This experiment applies to cases where the centralized server must collect expensive or time-consuming data. Here, the validation set now only allows 100 images, yet TAG prevents the backdoor attack and can achieve improved accuracy compared to the baseline robust aggregation methods. Hence, we additionally conclude we do not need validation data of the same quantity as other local users to discriminate between benign and malicious returning models.

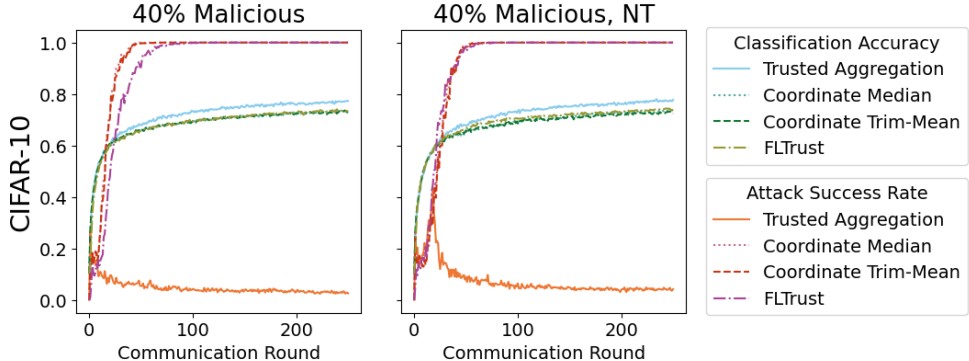

Figure 14: Model performance under DBA and Neurotoxin backdoor attacks with 40% malicious updates on CIFAR10 where the validation data set is 20% the size of the local users. Still, the proposed method TAG performs well, and the other three aggregation methods do not work well in preventing any backdoor attack.

## B    Extending Results To Imbalanced User Data Sets (Continued)

In this subsection, we provide additional figures that complement Section 4.4, to assistin understanding the effectiveness of backdoor attacks when local user data sets exhibit imbalanced distributions. With imbalanced data, we expect more variety in the changes a user can make to the output distributions. Intuitively, our defense should be less effective as malicious behavior should become more challenging to differentiate from benign behavior. This section's results help us understand how different backdoor defenses can be under other local user data distributions, TAG scaling coefficients, and the validation data distribution and size.

Figure 15 and Figure 16 show that even without tuning TAG's scaling coefficient $\theta$, our proposed defense is effective for imbalanced user data regardless of whether the trusted user has imbalanced data as well. Note that we are using $\theta = 2$ as obtained from tuning our defense method on balanced local user data sets for CIFAR10. Similar to other experimentation results, TAG is the only method to prevent our backdoor attacks without changing model performance for the original classification task.

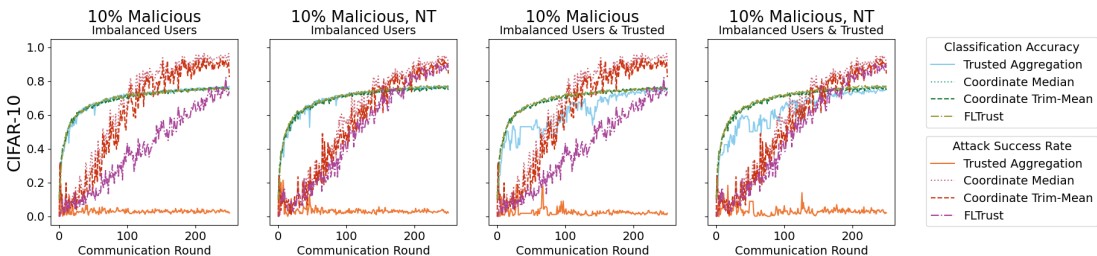

Figure 15: Model performance under DBA without and with Neurotoxin (NT) backdoor attacks with 10% malicious updates under imbalanced local data sets without tuned $\theta$. Column names indicate whether the trusted user (Trusted) is also imbalanced. The proposed method, TAG, performs well against backdoor attacks, even when the local user data sets are imbalanced. Again, the other defense methods do not prevent any backdoor attack under imbalanced data.

Consistently, TAG scaling is robust to changes under weaker attacks but needs application-specific tuning to offer its best backdoor defense. If $\theta$ is not modified from the previous experiments, all but one considered backdoor attacks are successful at 40% prevalence, see Figure 17. However, in Section 4.4, we observe that with proper tuning of $\theta$, even with imbalanced user data, TAG can prevent the powerful attack where 40% of returning user updates are malicious. TAG is a good choice for a defense method regardless of the data distribution of its users.

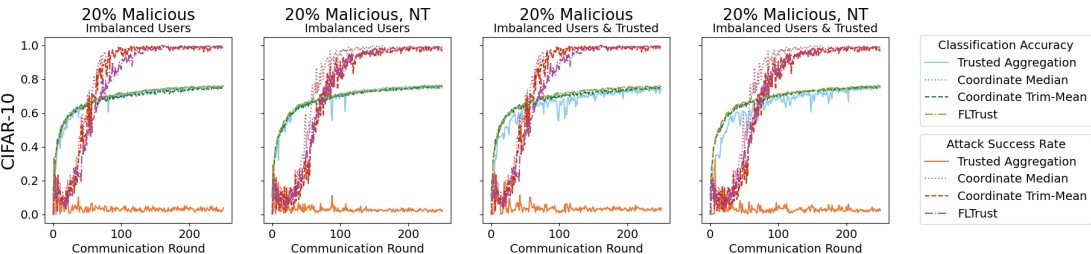

Figure 16: Model performance under DBA without and with Neurotoxin (NT) backdoor attacks with 20% malicious updates under imbalanced local data sets without tuned $\theta$. See above for further details.

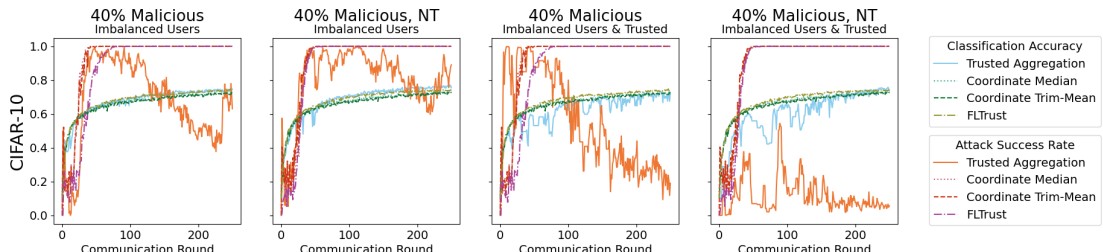

Figure 17: Model performance under DBA without and with Neurotoxin (NT) backdoor attacks with 40% malicious updates under imbalanced local data sets without tuned $\theta$. See above for further details.

### B.1 Discussion on Fairness

We use the imbalanced user experiments to understand our algorithm's fairness better. We do not wish to systematically exclude benign users from updating the model while defending against backdoor attacks. Here, we consider the CIFAR10 experiment with imbalanced users where 10% of model updates are malicious. When users have highly non-iid data, it becomes harder to understand whether that imbalance results from malicious behavior or natural heterogeneity. Similar results hold under different attack settings.

In Figure 18, we show that three users were excluded from participating often in federated training. The users with a low proportion of accepted updates to the shared model are labeled by their user ID for easy identification. While excluding the malicious user of the three is desired behavior, we systematically exclude two users as a consequence of filtering returning models. These users have an extremely high proportion of their local training data consisting of a single class label. As a result, their locally updated models have logits considered suspicious changes from the previous shared model, which should work well on all class labels.

We remark that when local datasets are highly heterogeneous, using a single shared model between users may not be appropriate. We leave a backdoor defense against highly heterogeneous users for future work. However, we expect the application of our method to have degraded performance as the degree of heterogeneity grows. We recommend the use of TAG when it is plausible that users share at least moderate similarities between their local datasets.

### B.2 Sensitivity of our Scaling Coefficient

In this section, we further study the sensitivity of our scaling hyper-parameter in the presence of imbalanced users on the higher-dimensional CIFAR100 data set. Recall, in Section 3 we suggest using the smallest scaling coefficient $\theta$ that results is desirable performance on the original task.

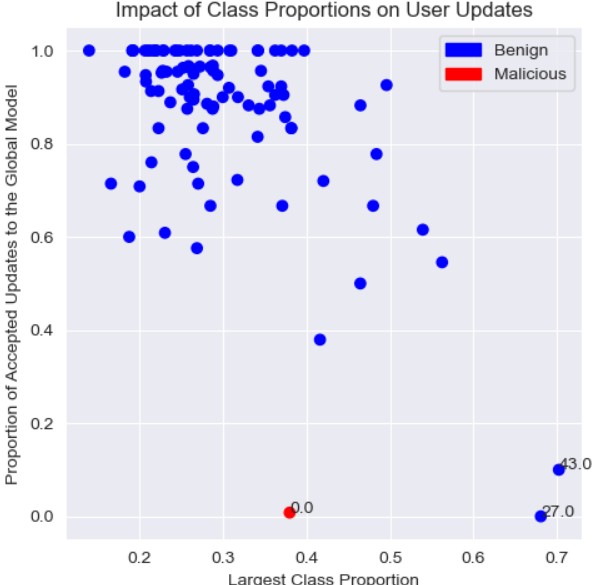

Figure 18: User participation under Neurotoxin backdoor attack with 10% malicious updates on CIFAR10. Across federated training, TAG frequently prevents three users from updating the global model. While excluding the malicious user of the three is desired behavior, as a consequence of thresholding model updates, we systematically exclude two users from participating in federated training. These other two users have an extremely high proportion of their local training data from a single class label, which results in suspicious changes to their locally updated model's logits.

Regarding hyperparameter selection, in Section 3, we recommend choosing the smallest scaling coefficient, $\theta$, that achieves desirable performance for classification accuracy. For example, tuning $\theta$ can be done from a simple grid search over select values. On CIFAR100, see Figure 19, classification accuracy becomes more variable round-to-round as $\theta$ decreases, and with $\theta = 1.1$, we see a notable change in main-task performance. Based on our recommendation, we would use the model created from $\theta = 1.25$.

Importantly, in Figure 19, we also observe that each of the scaling coefficients smaller than two ($\theta < 2$) prevented the DBA and Neurotoxin backdoor attacks with 10 and 20 percent malicious updates under imbalanced local data sets on CIFAR100. TAG's ability to prevent backdoor attacks across a range of $\theta$ is evidence of some robustness and stability to our hyperparameter choice. None of the attempted choices for $\theta$ were able to prevent the backdoor attack with 40 percent malicious updates. We believe it may be more difficult to defend against highly prevalent backdoor attacks with TAG for high-dimensional classification problems. Since our cutoff is based on the largest class change, with many classes, we would expect that even benign users may make a considerable change to at least one class.

We leave to future work to explore a strategy that does not rely on the maximum class distance to create a threshold. We believe that such a strategy will scale better to higher-dimensional classification problems. Also, it should better extend to distributional assumptions other than Uniform on the distances between logits between the previous global model and the locally updated copy of a benign user.

## C   Proof of Proposition 1

*Proof.* Let $v_c$ denote the distribution of distances between logits produced for class $c$ on a given data set by the previous global model and the locally updated copy of a benign user. Assume that each element has the Uniform distribution, $v_c \sim \text{Uniform}(0, b_c)$, from zero to some class-specific constant, $b_c$, for all benign users.

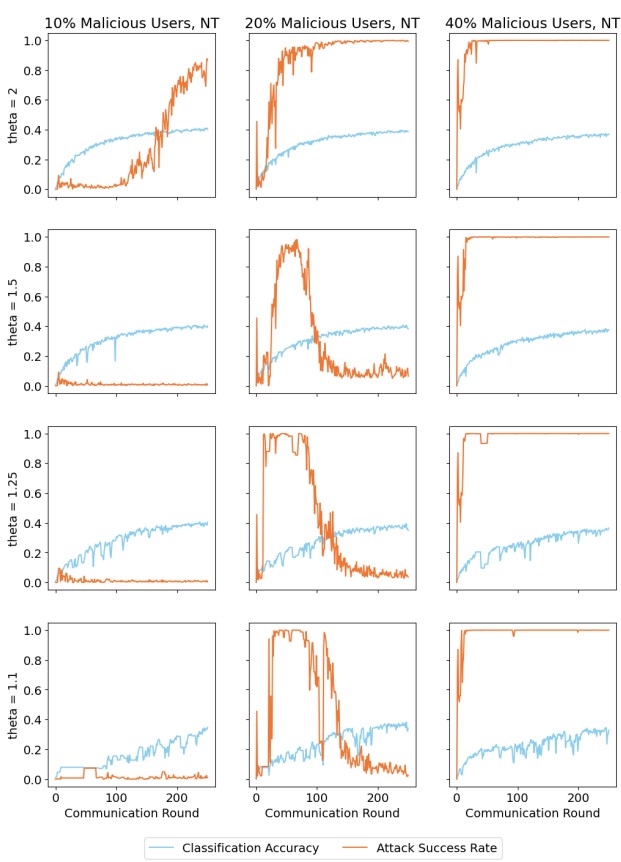

Figure 19: Model performance under DBA with Neurotoxin (NT) backdoor attacks with various percentages of malicious updates under imbalanced local data sets on CIFAR100. With scaling coefficients $\theta < 2$, TAG can prevent backdoor attacks with 10 and 20 percent malicious updates. With imbalanced local data sets on CIFAR100, TAG cannot prevent the backdoor attack against 40% malicious updates.

Define $v = \max_c [v_c]$ and $j = \arg \max_c [b_c]$.

$$v_j \leq v \Rightarrow \mathbb{E}[v_j] \leq \mathbb{E}[v]$$
$$\Rightarrow \frac{b_j}{2} \leq \mathbb{E}[v]$$
$$\Rightarrow b_j \leq 2\mathbb{E}[v]$$

$$v \leq b_j \Rightarrow \mathbb{E}[v] \leq \mathbb{E}[b_j]$$
$$\Rightarrow \mathbb{E}[v] \leq b_j$$

Therefore $\mathbb{E}[v] \leq b_j \leq 2\mathbb{E}[v]$. Moreover $\theta \times \mathbb{E}[v] = b_j$ for some $\theta \in [0, 1]$. $\qquad \square$

