# OpenReview forum: "Trusted Aggregation (TAG): Backdoor Defense in Federated Learning"
_TMLR — Accepted by TMLR_

### Review · Reviewer_vRTb · 2024-07-20

**Summary Of Contributions:**

The paper studies federated learning under backdoor attacks. The authors observe that the logits of models trained with and without backdoor attacks are different, and leverage this difference to propose a defense method that can prevent backdoor attacks while maintaining classification accuracy. The authors validate this method in a series of experiments by comparing it with other methods and showing robustness.

**Audience:**

Yes

**Claims And Evidence:**

No

**Requested Changes:**

Here are some other questions for improvement.

-  The atuhors observe that the distributions of model logits generated by malicious users, significantly differ from those produced by benign users. Is this specific to the dataset or the task in the paper?

-  Does the global model change over different rounds? In practice, the global model is not necessarily fixed. If it is changing based on current user models, how do you maintain a trust threshold?

- Any theoretical guarantees can be provided to justify performance in experiments?

- Why the robustness is stated for 40% malicious users? What happens if malicious users go above 40%?

**Strengths And Weaknesses:**

Strengths

- The authors propose a simple aggregation method that relies on the distribution difference of model logits from malicious and benign users. When a clean validation dataset is available, the authors show that this trusted dataset can be used to compute logits of a global model, determining a threshold for distinguishing malicious and benign users.

- To stabilize the threshold, the authors propose a global-min mean smoothing technique that averages top thresholds, which has a nice smoothing property preventing lucky malicious users.

- The authors demonstrate the effectiveness of the proposed method through a series of experiments, showing better performance than existing methods and robustness to changes in the dataset.

Weaknesses

- The proposed method relies on the distribution difference of model logits from malicious and benign users. However, this assumption might not always hold true. For instance, malicious users may target models other than classification models. Consequently, the proposed method may be limited to specific tasks.

- The proposed detection scheme is quite heuristic, and Proposition 1 assumes uniform logits, which are not realistic in practice. Conditions on the threshold should be examined for more realistic logits.

- There is no theoretical basis to justify the robustness of the proposed global-min mean smoothing scheme.

---

> ### Author Response · Authors · 2024-09-16
> **Official Comment by Authors**
>
> $\textbf{Remarks}$:
>
> The proposed method relies on the distribution difference of model logits from malicious and benign users. However, this assumption might not always hold true. For instance, malicious users may target models other than classification models. Consequently, the proposed method may be limited to specific tasks.
>
> The proposed detection scheme is quite heuristic, and Proposition 1 assumes uniform logits, which are not realistic in practice. Conditions on the threshold should be examined for more realistic logits.
>
> The authors observe that the distributions of model logits generated by malicious users, significantly differ from those produced by benign users. Is this specific to the dataset or the task in the paper?
>
> $\textbf{Reply}$:
>
> Thank you for pointing this out. We agree that the current algorithm is primarily designed to address backdoor robustness in classification tasks within federated learning. Extending this approach to other models, such as generative models in federated learning, is a promising direction for future research.
>
> Next, the uniform assumption of the logits may not apply to real-world scenarios. A better approach would be to allow the data to determine the scaling factor, i.e., selecting the scaling factor based on the distribution of logits observed in the experiments. In our experiments, we found that using a scaling factor of two, based on the uniform assumption, yielded good performance in most cases. As a result, we did not further explore data-dependent scaling factors. However, this is an excellent point; we discuss it in the updated paper.
>
> Regarding the observed distributional differences in logits, we provide additional evidence, see Figure 9, to demonstrate this phenomenon. The plots below show the distance scores (y-axis) generated by benign users (blue) and malicious users (red) across different training rounds (x-axis) on CIFAR10. The green curve represents the threshold generated by our proposed method. As can be seen, the red curve (malicious users) behaves very differently from the blue curves (benign users), which reflect the normal range of distance scores for benign users.
>
> In our original experiments, the three datasets used—standard computer vision datasets in federated learning—demonstrated consistent results. We expect our findings to generalize well to other datasets, although perhaps not to tasks beyond classification, as discussed earlier.
>
> $\textbf{Remark}$: “Does the global model change over different rounds? In practice, the global model is not necessarily fixed. If it is changing based on current user models, how do you maintain a trust threshold?”
>
> $\textbf{Reply}$:
>
> Yes, the global model changes with each round. Therefore, the threshold is not a fixed value but a dynamic one updated every round based on the distance between the global model's and the trusted model's logits. Since the trusted model is updated each round, this distance reflects the expected difference between a benign user's model and the global model from the previous round.
>
> $\textbf{Remark}$: “There is no theoretical basis to justify the robustness of the proposed global-min mean smoothing scheme. Any theoretical guarantees can be provided to justify performance in experiments?”
>
> $\textbf{Reply}$:
> Regarding our algorithm, we were unable to establish any theoretical guarantees. Therefore, we do not claim that our defense is certifiably robust. Instead, we evaluate the proposed method against strong attacks under strict conditions:
>
> 1. We assume that all malicious users are selected in each round. In real-world scenarios, users would be randomly selected to update the global model. We ensure that all malicious users are chosen to test whether the proposed method can successfully detect them.
>
> 2. We also allow the adversaries to coordinate their efforts and to constitute a high proportion of the total users.
> According to the 2024 survey paper on backdoor attacks in federated learning [1], these are strong assumptions that are unlikely to hold in practical settings.
>
> $\textbf{Remark}$: “Why the robustness is stated for 40% malicious users? What happens if malicious users go above 40%?”
>
> $\textbf{Reply}$:
>
> We acknowledge that the proposed method may only perform well if the proportion of malicious users is 40% or fewer. However, we emphasize that 40% is already an unusually high proportion in most real-world scenarios. If nearly half of the users are malicious, it becomes exceedingly challenging for training to succeed. Furthermore, many recent prominent works in this field [2-4] have used a maximum malicious user rate below 40%.

---

> ### Author Response · Authors · 2024-09-16
> **Official Comment by Authors - References**
>
> $\textbf{References}$:
>
> [1] Nguyen, Thuy Dung, et al. "Backdoor attacks and defenses in federated learning: Survey, challenges and future research directions." Engineering Applications of Artificial Intelligence 127 (2024): 107166.
>
> [2] Cao, Xiaoyu, et al. "Fltrust: Byzantine-robust federated learning via trust bootstrapping." Advances in Network and Distributed System Security Symposium (2021).
>
> [3] Zhang, Zaixi, et al. "Fldetector: Defending federated learning against model poisoning attacks via detecting malicious clients." Proceedings of the 28th ACM SIGKDD Conference on Knowledge Discovery and Data Mining. 2022.
>
> [4] Huang, Tiansheng, et al. "Lockdown: backdoor defense for federated learning with isolated subspace training." Advances in Neural Information Processing Systems 36 (2024).

---

> ### Comment · Reviewer_vRTb · 2024-09-26
>
> Thank you for the response. I don't have any further questions.

---

### Review · Reviewer_9GHt · 2024-08-31

**Summary Of Contributions:**

This paper introduces Trusted Aggregation (TAG), a robust defense mechanism against backdoor attacks in the federated learning framework. TAG mainly performs model filtering based on the differences in the logits of models trained with and without the presence of backdoor attacks. The proposed method demonstrates its effectiveness against backdoor attacks with multiple attackers, even in scenarios where 40% of clients are part of an attack.

**Audience:**

Yes

**Broader Impact Concerns:**

No concerns.

**Claims And Evidence:**

Yes

**Requested Changes:**

see the weakness above

**Strengths And Weaknesses:**

Strengths
- the motivation behind the proposed method is reasonable
- the experiments are sufficient to validate the effectiveness of the proposed method

Weaknesses
- the figures in the experiment are confusing, especially figure 4, different terms (accuracy or ASR) should be placed into different figures.
- the necessity of a clean data set to obtain a trusted user
- the sensitivity of the scaling coefficient

---

> ### Author Response · Authors · 2024-09-16
> **Official Comment by Authors**
>
> $\textbf{Remark}$: “the figures in the experiment are confusing, especially figure 4, different terms (accuracy or ASR) should be placed into different figures.”
>
> $\textbf{Reply}$:
>
> Thank you for your valuable feedback. We fully agree and have incorporated your suggestion to improve the clarity and readability of our key experimental figures. As recommended, we have separated the classification accuracy and attack success rate results into different plots. These adjustments can be seen in Figure 4 and Figure 8 of the updated paper.
>
> $\textbf{Remark}$: “the necessity of a clean data set to obtain a trusted user”
>
> $\textbf{Reply}$:
>
> We agree that the method would be even more effective if a clean dataset were not required. To address this concern, we tested the proposed method's performance when the size of the clean dataset is much smaller than a user’s dataset. The experimental results in Section D of the appendix demonstrate that TAG can defend against strong backdoor attacks with 40% malicious users even with a small, clean dataset. At the same time, other baseline methods fail, highlighting the advantage of using a clean dataset.
>
> The critical difference between TAG and the baseline methods is that our approach compares task performance based on model outputs. In contrast, based on update gradients, the baselines differentiate between malicious and benign users. We require both model weights and a clean set to compute the model outputs. In backdoor attacks, the loss typically combines the original task loss and the backdoor loss: $\text{Backdoor Loss} = \text{Original Task Loss} + \lambda \times \text{Backdoor Loss}$, where $\lambda$ is usually close to zero. As a result, the gradients of malicious users may appear similar to those of benign users, but our method can still detect them.
>
> However, the method would be more advantageous if a clean dataset were not required and performance could still be maintained. We consider this a promising area for future research.
>
> $\textbf{Remark}$: “the sensitivity of the scaling coefficient”
>
> $\textbf{Reply}$:
>
> Thank you for pointing this out. Regarding the sensitivity of our scaling coefficient, we made minimal effort to tune the hyperparameter. In most experiments, using a scaling coefficient of two without any tuning yielded excellent performance across datasets and attacks. Only in the case of STL10 and imbalanced user data did we need to fine-tune the scaling coefficient to achieve the best possible accuracy on the original classification task. In such cases, a simple grid search is sufficient to prevent backdoor attacks by identifying the smallest $\theta$ that achieves desirable classification accuracy.

---

> > ### Comment · Reviewer_9GHt · 2024-10-10
> >
> > Thank you for the response. I don't have any further questions.

---

### Review · Reviewer_qQGw · 2024-09-04

**Summary Of Contributions:**

The authors propose a new approach called TAG which prevents backdoor attacks from affecting the model while preserving the accuracy of the original classification task. TAG uses a small validation data set to estimate the maximum change a benign client's local training can impose on the shared model, which helps filter out malicious clients from updating the shared model.

The high level idea is as follows: in each communication round, a trusted user establishes a threshold for detecting malicious users by updating a copy of the global model using validation data, while other users train their local models. When models are returned by selected users, the logits for the validation data are generated and stored for comparison between the global, validation, and each user’s models. The distance between each user’s logits and the previous global model’s logits is then calculated for each class using a distributional difference function. This process creates a distance vector for each user, which determines which users can participate in the model update. The distance vector is thresholded to determine which users can participate in the update.
The authors also show that Global-Min Mean Smoothing (GMMS) combines the benefits of both a stable threshold in the long run, and a rapidly adjusting threshold earlier in training.

Experimental results on multiple data sets demonstrate that TAG effectively defends against backdoor attacks, even when up to 40% of the users are malicious on CIFAR10.

**Audience:**

Yes

**Claims And Evidence:**

Yes

**Requested Changes:**

See above.

**Strengths And Weaknesses:**

The techniques appear to perform very well in practice, for instance resisting the NT attack on VGG even when there are 40% of malicious users on CIFAR10. This is much better than some of the prior attacks proposed, which is a strong merit. This positive behavior continues even when certain parameters in the model are not optimally tuned on CIFAR10, and further more when the user data looks increasingly heterogenous.

On other datasets, I am curious how well the algorithm works in practice when $\theta$ is not tuned signficantly. While the experiments in the paper are promising, and show that in the absence of tuning, even up to 20% of malicious users can be resisted against the Neurotoxin attack, it is a bit unclear how things change when the dataset looks much different and more heterogenous, and the user does not have the benefit of tuning it in any manner (such as by looking at the case of balanced user data as is done for CIFAR10).

I am also curious about how the robustness of this algorithm interacts with fairness which is expected from the end-to-end models. I would be curious to see how the representation from underrepresented users clusters is affected by the thresholding - heterogeneity means that logits of certain users are likely not to look very similar to the global model from the previous round - these users would be thresholded away in many iterations, and the learnt model is not likely to treat all user clusters fairly.

I also believe that since the work does not prove a theorem about the nature of security offered against a class of attacks, it is more plausible that there are tailored attacks / adaptive attacks that can cause TAG to perform poorly. It's not clear how to reason about these attacks, and I would have appreciated to see some more discussion in this context, beyond the experiment in Fig. 7. In particular, are there settings where the algorithm performs ``worse'' than existing defenses against tailored attacks? If there does not appear to be, is it plausible that (under assumptions), the algorithm indeed resists any attack under a bound on the number of malicious users? While proving a theorem of this nature may be beyond the scope of the directions considered in the paper, it merits some discussion.

If the authors can address these comments satisfactorily, I am happy to see the paper accepted.

---

> ### Author Response · Authors · 2024-09-16
> **Official Comment by Authors - Part 1**
>
> We greatly appreciate your feedback and questions! You provided some excellent opportunities for further discussion, improving the quality of our work. In addition to clarifications, we have made several additions to the paper, which we summarize below.
>
> $\textbf{Remark}$: “On other datasets, I am curious how well the algorithm works in practice when theta is not tuned significantly. While the experiments in the paper are promising, and show that in the absence of tuning, even up to 20% of malicious users can be resisted against the Neurotoxin attack, it is a bit unclear how things change when the dataset looks much different and more heterogenous, and the user does not have the benefit of tuning it in any manner (such as by looking at the case of unbalanced user data as is done for CIFAR10).”
>
> $\textbf{Reply}$:
>
> Regarding the sensitivity of our scaling coefficient, we made minimal effort to tune the hyperparameter. In most experiments, using the default scaling coefficient, $\theta = 2$, yielded excellent performance across datasets and attacks without tuning.
>
> First, under the imbalanced user data setting for CIFAR10, the proposed method performs well without tuning when the proportion of malicious users is below 40%, as demonstrated in Figures 15 and 16 in the appendix. However, TAG faces challenges in the heterogeneous data setting when the proportion of malicious users reaches 40%, as shown in Figure 17. While TAG outperforms other defense methods in Figure 17, it does not consistently detect attacks. After tuning $\theta$, TAG regains its strong performance, as shown in Figure 6.
>
> Second, we have added a subsection B2, “Sensitivity of our Scaling Coefficient,” to our appendix. Here, we extend our experiments with imbalanced data to CIFAR100, a more challenging, high-dimensional classification problem. On CIFAR100, we can prevent backdoor attacks when the proportion of malicious updates is 10 or 20 percent. These results hold for various choices for the scaling coefficient, as shown in Figure 19. These findings indicate that it is not the sensitivity to our hyperparameter but the substantial proportion of malicious model updates that cause difficulties when local data sets are imbalanced.
>
> In conclusion, we acknowledge that many malicious users significantly impact the proposed method's performance under heterogeneous data settings. However, 40% of malicious users are uncommon in real-world scenarios. Addressing backdoor defenses under conditions of high malicious user proportions and heterogeneous data is an area we plan to explore in future work.
>
> $\textbf{Remark}$: “I am also curious about how the robustness of this algorithm interacts with fairness which is expected from the end-to-end models. I would be curious to see how the representation from underrepresented users clusters is affected by the thresholding - heterogeneity means that logits of certain users are likely not to look very similar to the global model from the previous round - these users would be thresholded away in many iterations, and the learnt model is not likely to treat all user clusters fairly.”
>
> $\textbf{Reply}$:
>
> Thank you for highlighting this critical issue. We have included an additional experiment to understand the trade-off between fairness and security better. Specifically, for the CIFAR-10 experiment with imbalanced users, we have prepared a plot displaying the number of times each user was allowed to contribute to the global model update. This visualization is presented in Figure 18 in the updated version of the paper. Three users exhibited relatively low participation rates in the model update process. Among these, one was identified as the attacker, while the other two users had over 70% of their data concentrated in a single class, representing an extreme case.
>
> We recognize that our detection mechanism may influence the algorithm's fairness, especially when deployed to counter backdoor attacks within heterogeneous data settings. We have incorporated an ablation study into the revised manuscript to enhance transparency regarding this trade-off between model security and fairness. This study is detailed in Section B.1, titled "Discussion on Fairness."

---

> ### Author Response · Authors · 2024-09-16
> **Official Comment by Authors - Part 2**
>
> $\textbf{Remark}$: “I also believe that since the work does not prove a theorem about the nature of security offered against a class of attacks, it is more plausible that there are tailored attacks / adaptive attacks that can cause TAG to perform poorly. It's not clear how to reason about these attacks, and I would have appreciated to see some more discussion in this context, beyond the experiment in Fig. 7. In particular, are there settings where the algorithm performs ``worse'' than existing defenses against tailored attacks? If there does not appear to be, is it plausible that (under assumptions), the algorithm indeed resists any attack under a bound on the number of malicious users? While proving a theorem of this nature may be beyond the scope of the directions considered in the paper, it merits some discussion.”
>
> $\textbf{Reply}$:
>
> Thank you for your insightful comments. We agree that the method cannot resist all types of attacks. The critical difference between TAG and the baseline methods is that while the baseline approaches differentiate between malicious and benign users based on update gradients, our approach compares task performance based on model outputs.
>
> For backdoor attacks, the loss is typically a combination of the original task loss and the backdoor loss: $\text{Backdoor Loss} = \text{Original Task Loss} + \lambda \times \text{Backdoor Loss}$, where $\lambda$ is usually close to zero. As a result, the gradients of malicious users may appear similar to benign users, but our method can still detect them.
>
> Different local minima can produce comparable model outputs in highly non-convex loss landscapes. In such cases, our method may be less effective at filtering out models that are more easily detected by gradient-based defenses, highlighting the importance of using multiple defenses with diverse detection strategies to improve security against backdoor attacks. A promising future direction would be exploring combining these strategies to enhance models' robustness against targeted backdoor attacks. We have included this discussion in the updated version of the paper along with our main experimental results in Section 4.2, “Comparison of Defense Methods Against Backdoor Attacks.”

---

### Author Response · Authors · 2024-09-16
**Official Comment by Authors**

We would like to thank the reviewers for their most valuable comments. We have responded to all the review comments in the latest version of the paper. Our responses are detailed below, point by point. Comments from the reviewers are listed in italics, while our responses are in standard font. For easy referencing, changes in the revision are highlighted in $\textcolor{blue}{blue}$.

---

### Decision · Action_Editor_oeDQ · 2024-11-05

**Recommendation:** Accept as is

**Comment:**

This paper proposes a new approach called TAG which prevents backdoor attacks from affecting the model while preserving the accuracy of the original classification task. TAG uses a small validation data set to estimate the maximum change a benign client's local training can impose on the shared model, which helps filter out malicious clients from updating the shared model. The motivation behind the proposed method is reasonable. More importantly, to stabilize the threshold, the authors propose a global-min mean smoothing technique that averages top thresholds, which has a nice smoothing property preventing lucky malicious users. The authors also demonstrate the effectiveness of the proposed method through a series of experiments, showing better performance than existing methods and robustness to changes in the dataset. The authors address reviewers' concerns well in their rebuttal. Thus, I would like to recommend accept as is.

**Audience:**

Yes

**Claims And Evidence:**

Yes